# Pentacyclic Triterpenoids from *Sabia discolor* Dunn and Their *α*-Glycosidase Inhibitory Activities

**DOI:** 10.3390/molecules27072161

**Published:** 2022-03-27

**Authors:** Jin-Hong Ma, Dan Hu, Lu-Lu Deng, Jiang Li, Xiao-Jiang Hao, Shu-Zhen Mu

**Affiliations:** 1State Key Laboratory of Functions and Applications of Medicinal Plants, Guizhou Medical University, Guiyang 550014, China; jinhongma2022@163.com (J.-H.M.); danhu1996@163.com (D.H.); luludeng@gzcnp.cn (L.-L.D.); jiangli@gzcnp.cn (J.L.); haoxj@mail.kib.ac.cn (X.-J.H.); 2The Key Laboratory of Chemistry for Natural Products of Guizhou Province and Chinese Academy of Sciences, Guiyang 550014, China; 3School of Pharmacy, Guizhou Medical University, Guiyang 550025, China; 4Kunming Institute of Botany, Chinese Academy of Sciences (CAS), Kunming 650201, China

**Keywords:** *Sabia discolor* Dunn, pentacyclic triterpenoids, isolation and purification, *α*-glycosidase inhibitory activities

## Abstract

Four new pentacyclic triterpenoids named Sabiadiscolor A–D (**1** and **7**–**9**) together with eleven known ones were isolated by repeated column chromatography. Their structures were identified and characterized by NMR and MS spectral data as 6 oleanane-type pentacyclic triterpenoids (**1**–**6**), 7 ursane-type ones (**7**–**13**), and 2 lupanane-type ones (**14**–**15**). Except for compound **15**, all other compounds were isolated from *Sabia discolor* Dunn for the first time. Their *α*-glycosidase inhibitory activities were evaluated, which showed that compounds **1**, **3**, **8**, **9**, **13**, and **15** implied remarkable activities with IC_50_ values ranging from 0.09 to 0.27 μM, and the preliminary structure–activity relationship was discussed.

## 1. Introduction

Diabetes mellitus (DM) is one of the most prevalent metabolic diseases worldwide. This disease is a chronic metabolic disease mainly characterized by hyperglycemia caused by a variety of factors, among which type 2 diabetes mellitus (T2DM) is the most common, accounting for 90% of the total number of diabetic patients. T2DM is a type of diabetes that is associated with an imbalance in glucagon/insulin homeostasis that leads to the formation of amyloid deposits in the brain, in pancreatic islet cells, and possibly in the kidney glomerulus. With increasing human material resources and improvements in living standards, the incidence of T2DM is increasing, which seriously affects human health and quality of life. When diet and exercise fail to control hyperglycemia, patients are forced to start therapy with antidiabetic agents. Currently, long-term medication remains an important tool for T2DM control and treatment, and these drugs are characterized by low bioavailability and immediate drug release, resulting in the need to increase the frequency of administration to achieve therapeutic goals. It is inconvenient for the patient [1]. Therefore, there is no ideal drug for the treatment of this disease and it is still urgent and necessary to develop new candidates with improved clinical therapeutic effects. Natural products, particularly those derived from plants, have been proven to exert anti-diabetic effects via diverse mechanisms [2,3]. However, these drugs present several drawbacks that can affect the course of treatment. *α*-Glucosidase inhibitors are an important class of drugs that can be used for the treatment of T2DM and widely exist in fruits, leaves, seeds, and other tissues and organs of plants. In the past 30 years, research on *α*-GI from Chinese herbal medicine has become active at home and abroad, and has gradually become a hot spot in the prevention and treatment of diabetes [4]. Based on this situation, in our continuous discovery of structurally interesting and biologically active triterpenes from medicinal plants [5,6,7], four new pentacyclic triterpenes (**1** and **7**–**9**), including one new oleanane and three new ursane-type triterpenes, as well as eleven known triterpenes, were isolated from the dried stems of *S. discolor* Dunn. Furthermore, the *α*-glycosidase inhibitory activities of these fifteen triterpenoids were screened by an enzyme-inhibitor model using maltose as a substrate. Herein, we mainly describe the structural elucidation of four new pentacyclic triterpenes (**1** and **7**–**9**) and the *α*-glycosidase inhibitory activities of all triterpenoids obtained from *S. discolor* Dunn. It will be of great significance to provide a scientific basis for the utilization and development of plant resources of genus *Sabia*.

## 2. Results and Discussion

### 2.1. Structural Analysis of New Compounds

The crude petroleum ether extract of *S. discolor* Dunn was isolated and purified by various column chromatography techniques, including MCI gel, Sephadex LH-20, silica gel, RP-C18 silica gel, and a semipreparative HPLC column, allowing for the isolation of four new pentacyclic triterpenes, namely Sabiadiscolor A–D (**1** and **7**–**9**), along with eleven known compounds. Their structures are shown in Figure 1. Compared with the literature, eleven of these structures were known compounds based on their NMR and MS data, and were identified as ursolic acid (**2**) [8], juglangenin A (**3**) [9], 3*β*, 28-dihydroxy-12-oleanene-l-one (**4**) [10], 3-hydroxyolean-12-en-1-one (**5**) [11], 1*α*, 2*α*, 3*β*-trihydroxyl-olean-12-en-28-oic acid (**6**) [12], dandelion alkan-3*β*, 20*β*-diol (**10**) [13], olean-12-ene-1, 3-diol (**1****1**) [14], 3-oxo-20S-hydroxytaraxastane (**12**) [15], *ψ*-taraxasterone (**13**) [16], betulinic acid (**14**) [17], and birch ester alcohol (**1****5**) [18].

Compound **1** was obtained as a white solid and its molecular formula was inferred to be C_30_H_48_O_3_ by HR-ESI-MS with *m*/*z* 457.3671 [M + H]^+^ (calc. 457.3676). The IR spectrum showed absorption bands for the presence of a hydroxy group (3477 cm^−1^) and a ketone carbonyl group (1713 cm^−1^). ^1^H NMR (Table 1 and Appendix A), ^13^C NMR (Table 2 and Appendix A), and DEPT (Appendix A) spectral data revealed the presence of eight quaternary carbons, five methine groups, ten methylene groups, and seven methyl groups, including a carbonyl group (*δ*_C_ 214.6) and a trisubstituted olefinic unit (*δ*_C_ 122.9 (CH) and *δ*_H_ 5.20 (dd, *J* = 4.5, 2.9 Hz, 1H); *δ*_C_ 144.1 (C)). According to ^1^H NMR (Table 1) spectral data, the compound has seven methyl groups at *δ*_H_ 0.88 (3H, s, H-30), 0.90 (3H, s, H-29), 1.00 (3H, s, H-26), 1.03 (3H, s, H-24), 1.10 (3H, s, H-23), 1.21 (3H, s, H-27), and 1.32 (3H, s, H-25). The comparison of the NMR data of compound **1** with those of the known compound **4** [10] in Table 1 and Table 2 showed that the two compounds should share the same basic skeleton and that both were very similar. According to further HMBC (Appendix A) correlations in Figure 2A, H-3 (*δ*_H_ 3.88, m, 1H) was correlated with C-1 (*δ*_C_ 214.6), C-5 (*δ*_C_ 51.3), and C-24 (*δ*_C_ 22.3), and H-28 (*δ*_H_ 3.30 and 3.21, 2H) was correlated with C-16 (*δ*_C_ 21.9), C-17 (*δ*_C_ 36.9), and C-18 (*δ*_C_ 42.5). Based on this data, it was predicted that the two hydroxyl groups should be at positions C-3 and C-28. Therefore, the planar structure of compound **1** was the same as that of compound **4**, as shown in Figure 2A. The relative configuration of compound **1** was further determined according to the NOESY (Appendix A) correlation spectrum in Figure 2B. The correlation signals between H-23, H-25, and H-3 indicate that H-3 is in the *β* configuration and the 3-substituted hydroxyl group has an *α* configuration. The relative configuration of compound **4** was determined according to the NOESY (Appendix A) correlation spectrum in Figure 2C. The correlation signals between H-24 and H-3 indicated that H-3 was the *α* configuration and the 3-substituted hydroxyl group has a *β* configuration. Therefore, it was confirmed that compound **1** and compound **4** are isomers, and compound **1** is 3*α*, 28-dihydroxy-12-oleanene-l-one, named Sabiadiscolor A.

Compound **7** was obtained as a white solid and its molecular formula was inferred to be C_30_H_50_O_2_ with 6 degrees of unsaturation by HR-ESI-MS with 465.3701 [M + Na]^+^ (calc. 465.3703). The IR spectrum of this compound revealed the presence of a hydroxyl group (3368 cm^−1^). As shown in Table 1 and Table 2, its ^1^H NMR (Appendix A), ^13^C NMR (Appendix A), and DEPT (Appendix A) spectral data showed that compound **7** contained thirty carbons, including six quaternary carbons, eight methine groups, eight methylene groups, and eight methyl groups. ^1^H NMR (Table 1) spectral data at *δ*_H_ 0.80 (3H, s, H-28), 0.99 (3H, s, H-30), 1.01 (3H, s, H-27), 1.09 (3H, s, H-24), 1.17 (3H, s, H-26), 1.24 (3H, s, H-23), 1.26 (3H, s, H-25), and 1.65 (3H, s, H-29) showed seven methyl groups at the sp^3^ quaternary carbons and one methyl group at the sp^3^ tertiary carbon. According to the above data, compound **7** was inferred to be a five-membered ring triterpene with one olefinic unit (*δ*_C_ 140.0 (C); *δ*_C_ 119.3 (CH) and *δ*_H_ 5.33 (dd, *J* = 6.7, 2.0 Hz, 1H)) and two oxygen-substituted methine groups (*δ*_C_ 79.7 (CH) and *δ*_H_ 3.80 (dd, *J* = 11.1, 4.2 Hz, 1H); *δ*_C_ 75.5 (CH) and *δ*_H_ 3.63 (dd, *J* = 12.0, 4.4 Hz, 1H)). Compared with the chemical structure of the pentacyclic triterpenes isolated from the genus *Sabia*, all spectral data showed that the compound was an ursane-type pentacyclic triterpene and was structurally similar to the known compound, which is 20-taraxastene-3*β*, 22*α*-diol [19]. By further analyzing the HSQC (Appendix A) and HMBC (Appendix A) correlation signals of compound **7** in Figure 3A, it was shown that H-3 (*δ*_H_ 3.63, dd, *J* = 12.0, 4.4 Hz) was correlated with C-2 (*δ*_C_ 39.5) and C-24 (*δ*_C_ 16.2), while H-1 (*δ*_H_ 3.63, dd, *J* = 12.0, 4.4 Hz) was correlated with C-3 (*δ*_C_ 75.5), C-5 (*δ*_C_ 52.1), and C-25 (*δ*_C_ 13.2). Based on the above-mentioned data, it was predicted that the two hydroxyl groups should be at positions C-1 and C-3. According to the ^1^H-^1^H COSY (Appendix A) signal in Figure 3A, both H-3 (*δ*_H_ 3.63) and H-1 (*δ*_H_ 3.80) are correlated with H-2 (*δ*_H_ 2.36), as shown in Figure 3A. The relative configuration of compound **7** was further determined according to the NOESY (Appendix A) correlation spectrum in Figure 3B and Appendix A. The correlation signals between H-3 and H-24 indicate that H-3 has an *α* configuration. The correlation signals between H-3 and H-1 indicate that H-1 has an *α* configuration. Both substituted hydroxyl groups at C-1 and C-3 had a *β* configuration. Therefore, the structure of the compound can be determined as 20-taraxastene-1*β*, 3*β* -diol, named Sabiadiscolor B.

Compound **8** was obtained as a white solid and its molecular formula was inferred to be C_30_H_50_O_2_ by HR-EI-MS with *m/z* 442.3806 [M]^+^. The IR spectrum absorption at 3413 cm^−1^ revealed the presence of the hydroxyl group. In Table 1 and Table 2, the ^1^H NMR (Appendix A), ^13^C NMR (Appendix A), and DEPT (Appendix A) spectral data of compound **8** showed that it also contained 30 carbons, including six quaternary carbons, six methine groups, and ten methylene and eight methyl groups. Two sp^3^ methines (*δ*_C_ 77.9 (CH) and *δ*_H_ 3.48 (m, 1H); *δ*_C_ 75.0 (CH) and *δ*_H_ 3.26 (dd, *J* = 12.4, 4.3 Hz, 1H)) were typical of oxygen-bearing groups. Similar to compound **7**, eight methyl groups at *δ*_H_ 0.74 (3H, s, H-28), 0.76 (3H, s, H-27), 0.93 (3H, s, H-23), 0.96 (3H, s, H-24), 0.96 (3H, s, H-25), 0.99 (3H, s, H-29), 1.06 (3H, s, H-26), and 1.57 (3H, s, H-30), and one trisubstituted olefinic unit (*δ*_C_ 118.8 (CH) and *δ*_H_ 5.30 (s, 1H); *δ*_C_ 139.9 (C)) existed in compound **8**. According to NMR spectral data, compound **8** was found to be very similar to the known compound pseudotaraxasterol [20]. By comparing their NMR data, it was found that the main difference lies in the chemical shift values of C-6, C-7, C-23, and C-25. Therefore, it was speculated that another hydroxyl group might be at C-6 or C-7. The HSQC (Appendix A) and HMBC (Appendix A) spectra of compound **8** showed that H-3 (*δ*_H_ 3.50) was correlated with C-1 (*δ*_C_ 38.1) and C-23 (*δ*_C_ 12.0). H-6 (*δ*_H_ 3.29) is associated with C-7 (*δ*_C_ 17.9), C-24 (*δ*_C_ 27.8), and C-26 (*δ*_C_ 16.3). The relative configuration of compound **8** was determined according to its NOESY (Appendix A) correlation spectrum, as shown in Figure 4B. The strong correlations of Me-23/H-3/H-5 and H-6/Me-23 indicated that these protons or methyl groups were α-orientations. Thus, Me-23, H-3, H-5, and H-6 were arbitrarily assigned *α*-orientations, while two hydroxyl groups on the third and sixth carbons were *β*-oriented. Therefore, the structure of compound **8** can be determined to be 6*β*-pseudotaraxasterol, named Sabiadiscolor C.

Compound **9** was obtained as a white solid and its molecular formula was inferred to be C_30_H_52_O_2_ by HR-ESI-MS with *m/z* 443.3339 [M − H]^−^. The IR spectrum revealed the presence of the hydroxyl group (3388 cm^−1^). All 30 carbons observed in the ^1^H NMR (Appendix A), ^13^C NMR (Appendix A), and DEPT (Appendix A) spectral data could be classified into six sp^3^ quaternary carbons, eight sp^3^ methine groups, eight sp^3^ methylene groups, and eight methyl groups, as shown in Table 1 and Table 2. Among them, one sp^3^ methine (*δ*_C_ 77.9 (CH) and *δ*_H_ 3.49 (dd, *J* = 10.6, 5.6 Hz, 1H)) and one sp^3^ quaternary carbon(*δ*_C_ 73.8 (C)) were ascribed as bearing oxygen atoms. According to ^1^H NMR data (Table 1), the compound has eight methyl groups at *δ*_H_ 0.90 (3H, s, H-24), 1.01 (3H, s, H-28), 1.03 (3H, d, H-29), 1.03 (3H, d, H-30), 1.08 (3H, s, H-25), 1.27 (3H, s, H-23), 1.33 (3H, s, H-26), and 1.42 (3H, s, H-27). The comparison of the NMR data of compound **9** with the known compound ursan-3*β*, which is 5*α*-diol [21], suggested that compound **9** possessed an ursane-type pentacyclic triterpene skeleton. The HSQC (Appendix A) and HMBC (Appendix A) spectra of compound **9** showed that C-13 (*δ*_C_ 73.8) was correlated with H-14 (*δ*_H_ 1.42) and H-18 (*δ*_H_ 1.33), which revealed that the hydroxyl group should be assigned at C-13. The relative configuration of compound **9** was determined according to its NOESY (Appendix A) correlation spectrum, as shown in Figure 5B. The strong correlations of Me-24/H-3/H-5 indicated that Me-24, H-3, and H-5 were *α*-orientations, while the hydroxyl group on the third carbon was *β*-oriented. Therefore, the structure of compound **9** can be determined as ursan-3*β*, 13*β*-diol, named Sabiadiscolor D.

### 2.2. α-Glycosidase Inhibitory Activities

All compounds (**1**–**15**) isolated from *S. discolor* Dunn were evaluated for their *α*-glycosidase inhibitory activity. As shown in Table 3, compounds **1**, **3**, **8**, **9**, **13**, and **15** showed remarkable activities with IC_50_ values from 0.09 to 0.27 μM, while compound **7** showed weak activity with an IC_50_ value of 0.56 ± 0.0331 μM. The other compounds had low inhibitory activity against *α*-glycosidase and are not listed in Table 3.

### 2.3. Discussion

By modern natural medicinal chemistry experiments, fifteen natural pentacyclic triterpenoids (**1**–**15**) were obtained and identified from the traditional Chinese ethnic medicinal plant named *S. discolor* Dunn, collected from minority areas, and four of them (**1** and **7**–**9**) were new compounds. Their preliminary *α*-glycosidase inhibitory activities were evaluated. The results showed that six compounds (**1**, **3**, **8**, **9**, **13**, and **15**) showed remarkable activities with IC_50_ values of 0.27 ± 0.0499, 0.11 ± 0.0222, 0.23 ± 0.0135, 0.23 ± 0.0307, 0.26 ± 0.0383, and 0.09 ± 0.0045 μM, respectively. It was found that ursane-type pentacyclic triterpenes have better hypoglycemic activities and especially new compounds **1, 8**, and **9** have more significant activity than the positive control (Acarbose), which revealed that they might be a class of potential *α*-glycosidase inhibitors. According to the structure and activity data of these pentacyclic triterpenoids with potential hypoglycemic activity, it is speculated that the number and location of hydroxyl groups as well as double-bond groups might contribute more greatly to the inhibition rate of *α*-glycosidase, and both oleanane-type pentacyclic triterpenoids and lupanane-type ones should be the potential *α*-glycosidase inhibitor. Due to limited quantity of the isolated compounds, it is not possible to systematically discuss the structure–activity relationship of such compounds, but compound **15** with the lupanane-type pentacyclic triterpenoid skeleton was implied to have the best activity.

*S. discolor* Dunn is one of the most important species in the genus *Sabia*, which is rich in resources in the minority areas of southwest China. The medical plant is used to treat rheumatism, bone pain, bruises, hepatitis, and other diseases in the folk [22]. The main types of chemical constituents in the genus *Sabia* included pentacyclic triterpenoids, alkaloids, benzene derivatives, and fatty acids [23], but there was less literature reporting on the chemical composition of *S. discolor* Dunn. As our research suggested its hypoglycemic activity for the first time, it provided an important basis for the comprehensive utilization of this plant resource.

## 3. Materials and Methods

### 3.1. General Experimental Materials

One-dimensional and 2D NMR spectra were measured on a Bruker AM-600 spectrometer. HRESIMS data were obtained by Q EXACTIVE FOCUS (Thermo Fisher Technologies Co. Ltd., Waltham, MA, USA) spectrometers. Electrospray ionization (ESI) data were obtained by an HP 1100SMD. Preparative HPLC separations were run on a SEP system (Beijing Sepuruisi Scientific Co., Ltd., Beijing, China) equipped with a variable-wavelength UV detector using a YMC-Pack ODS-A column (250 × 20 mm, 5 µm). IR spectra were obtained using a Bruker Tensor-27 instrument (Bruker, Munich, Germany). The extract was obtained through a 300 L extraction tank (JF21060, Jiangsu Jufeng Machinery Co. Ltd., Huaian, China). Column chromatography (CC) was performed on silica gel (40–80 mesh, 200–300 mesh, and 300–400 mesh, Qingdao Haiyang Chem. Ind. Ltd., Qingdao, China), silica gel H (40–80 μm mesh, Qingdao, China), Sephadex LH-20 (40–70 μm, Amersham Pharmacia Biotech AB, Uppsala, Sweden), MCI gel (CHP20P, 75–150 μm, Mitsubishi Chemical Industry Co. Ltd., Kenyworth, NJ, USA), and C18 reversed-phase silica gel (20–45 μm, Merck, Darmstadt, Germany). TLC plates were precoated with silica gel GF254 (Qingdao Haiyang Chem. Ind. Ltd., Qingdao, China). All solvents were of analytical grade (Anergy Chemical Reagents Co. Ltd., Shanghai, China; Shanghai Titan Technology Co. Ltd., Shanghai, China).

Altogether, dimethyl sulfoxide (DMSO; Beijing Mr. Lai Treasure Company, Beijing, China); phosphate buffer solution (PBS; HyClone); centrifuge tubes, 96 cell culture plates, and other consumables (NEST Biotechnology Co. Ltd., Beijing, China); and *α*-glucosidase (Sigma Company, St. Louis, MO, USA) from saccharomyces cerevisiae were obtained for this study. Acarbose (Shanghai Yuanye Biotechnology Co. Ltd., Shanghai, China) was used as a positive control. The absorbance was read using a microplate reader (Varioskan LUX, Thermo, Waltham, MA, USA) at 405 nm. The results were obtained for at least three independent experiments.

### 3.2. Plant Materials

*S. discolor* Dunn was collected from Zhutou Mountain, Guangxi Province. The dried stems and leaves were identified by Professor Qing-Wen Sun of Guizhou University of Traditional Chinese Medicine. The samples were stored in the Key Laboratory of Chemistry for Natural Products, Chinese Academy of Sciences, Guizhou Province.

### 3.3. Extraction and Isolation

The dry, powdered stems of *S. discolor* Dunn (18 kg) were refluxed three times with 75% ethanol for 4, 3, and 3 h successively. The ethanol in the extract was fully recovered and then the alkaline substances in the extract were removed by 10% bitartrate acidification to pH = 2. The above extracts were extracted and recovered with petroleum ether to obtain 120 g of extract. The petroleum ether extract was subjected to a petroleum ether/ethyl acetate solvent gradient (80:1–1:1) by silica gel CC to obtain nine fractions (Fr. 1–Fr. 9). The separation process is shown in Figure 6.

Fr. 4 was subjected to silica gel CC (petroleum ether/methylene dichloride, 8:1) to obtain compound **14** (84 mg). Fr. 5 was subjected to Sephadex LH-20 CC (eluted with CHCl_3_/MeOH, 1:1) to obtain compound **13** (5 mg). Fr. 6 was subjected to Sephadex LH-20 CC (eluted with CHCl_3_/MeOH, 1:1) to obtain four subfractions (Fr. 6.1–Fr. 6.4) and Fr. 6.2 was subjected to silica gel column (petroleum ether/ethyl acetate, 10:1) to obtain five fractions (Fr. 6.2.1–Fr. 6.2.5). Compound **5** (6 mg) was obtained by Sephadex LH-20 CC (eluted with CHCl_3_/MeOH, 1:1) from Fr. 6.2.1. Compounds **3** (95 mg) and **1** (105 mg) were obtained by silica gel CC (petroleum ether/diethylamine, 15:1) from Fr. 6.2.2. Compound **12** (28 mg) was obtained by Sephadex LH-20 CC (eluted with MeOH) from Fr. 6.2.5. Fr. 7 was subjected to Sephadex LH-20 CC (eluted with CHCl_3_/MeOH, 1:1) to obtain eight subfractions (Fr. 7.1–Fr. 7.8) and Fr. 7.4 was purified by both Sephadex LH-20 CC (eluted with MeOH) and silica gel CC (petroleum ether/ethyl acetate, 6:1) to obtain compound **15** (16 mg). Fr. 8 was filtered and subjected to silica gel CC (petroleum ether/diethylamine, 15:1) to afford compound **7** (28 mg). The residue was subjected to Sephadex LH-20 CC (eluted with MeOH) to obtain five fractions (Fr. 8.1–Fr. 8.5). Fr. 8.3 was subjected to silica gel CC (petroleum ether/ethyl acetate, 3:1) to obtain compound **4** (40 mg). Compound **11** (93 mg) was obtained by Sephadex LH-20 CC (eluted with CHCl_3_/MeOH, 1:1) from Fr. 4. Compound **10** (9 mg) was obtained by silica gel CC (petroleum ether/chloroform, 100:1) from Fr. 8.6. Fr. 9 was subjected to Sephadex LH-20 CC (eluted with CHCl_3_/MeOH, 1:1) to obtain four subfractions (Fr. 9.1–Fr. 9.4) and Fr. 9.1 was subjected to Sephadex LH-20 CC (eluted with MeOH) to afford compounds **6** (34 mg) and **9** (9 mg). Fr. 9.4 was subjected to a decompression silica gel CC (petroleum ether/chloroform, 50:1) to afford compounds **8** (11 mg) and **2** (15 mg).

### 3.4. Assay of α-glycosidase Inhibition

The inhibitory activity of *α*-glucosidase was determined by the PNPG method [24]. PBS, different concentrations of samples or positive drugs, *α*-glucosidase, and 10% DMSO solution (100 μL) were mixed by shaking and incubated for 15 min in a constant temperature incubator at 37 °C. Then, 20 μL of PNPG (2.5 mmol/L) was added, mixed by shock, and incubated in a 37 °C constant-temperature incubator for 15 min. Then, 80 μL of Na_2_CO_3_ (0.8 mmol/L) solution was added to stop the reaction and the absorbance was measured by a microplate reader at 405 nm. Five groups were established, including the blank group (90 μL PBS + 10 μL 10% DMSO + 20 μL PNPG (2.5 mmol/L) + 80 μL Na_2_CO_3_ (0.8 mmol/L)), background group (90 μL PBS + 10 μL compounds + 20 μL PNPG (2.5 mmol/L) + 80 μL Na_2_CO_3_ (0.8 mmol/L)), negative control group (70 μL PBS + 20 μL *α*-glucosidase + 10 μL 10% DMSO + 20 μL PNPG (2.5 mmol/L) + 80 μL Na_2_CO_3_ (0.8 mmol/L)), positive control group (70 μL PBS + 10 μL acarbose + 20 μL *α*-glucosidase + 20 μL PNPG (2.5 mmol/L) + 80 μL Na_2_CO_3_ (0.8 mmol/L)), and drug administration group (70 μL PBS + 20 μL *α*-glucosidase + 10 μL compounds + 20 μL PNPG (2.5 mmol/L) + 80 μL Na_2_CO_3_ (0.8 mmol/L)), with 3 parallel replicates in each group.

Inhibition rate = [(OD _negative_ − OD _blank_) − (OD _sample_ − OD _background_)]/(OD _negative_ − OD _blank_) × 100%.

If the inhibitory rate was close to or higher than that of acarbose, the compound was considered to have *α*-glycosidase inhibitory activity. The IC_50_ of the potential compound was measured and calculated by the same method after 5-fold dilution.

## 4. Conclusions

To isolate *α*-glycosidase inhibitors from natural products, almost all fifteen pentacyclic triterpenoids were isolated for the first time from the branches and leaves of *S. discolor* Dunn, which is a kind of ethnic medicinal plant. These triterpenoids included six oleanane-type pentacyclic triterpenoids, seven ursane-type triterpenoids, and two lupanane-type triterpenoids. Among them, four compounds (**1** and **7**–**9**) were new pentacyclic triterpenoids. By further evaluating the *α*-glycosidase inhibitory activities of these compounds, compounds **1**, **3**, **8**, **9**, **13**, and **15** showed remarkable activities with IC_50_ values of 0.27 ± 0.0499, 0.11 ± 0.0222, 0.23 ± 0.0331, 0.23 ± 0.0307, 0.26 ± 0.0383, and 0.09 ± 0.0045 μM, respectively. The results revealed that pentacyclic triterpenoids could be a class of potential *α*-glycosidase inhibitors.

## Figures and Tables

**Figure 1 molecules-27-02161-f001:**
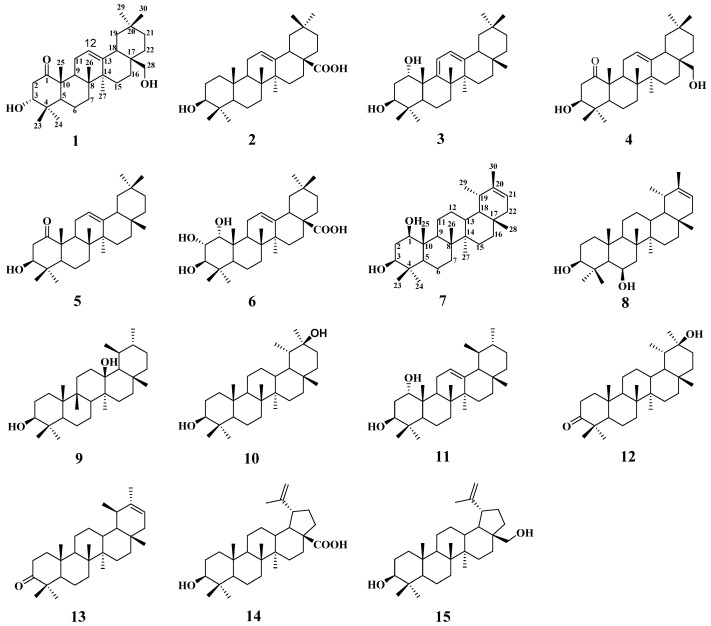
Structures of compounds **1**–**15**.

**Figure 2 molecules-27-02161-f002:**
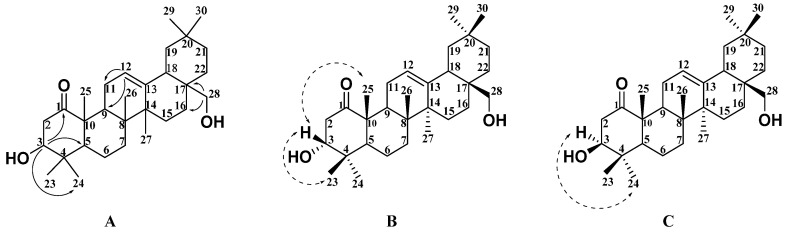
Key HMBC (plain arrow; (**A**)), NOESY (dash arrow) correlations of compound **1** (**B**), and NOESY (dash arrow) correlations of compound **4** (**C**).

**Figure 3 molecules-27-02161-f003:**
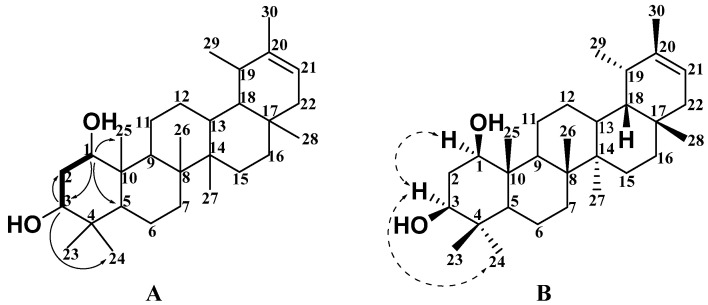
Key H-H COSY (bold), HMBC (plain arrow; (**A**)), and NOESY (dash arrow; (**B**)) correlations of compound **7**.

**Figure 4 molecules-27-02161-f004:**
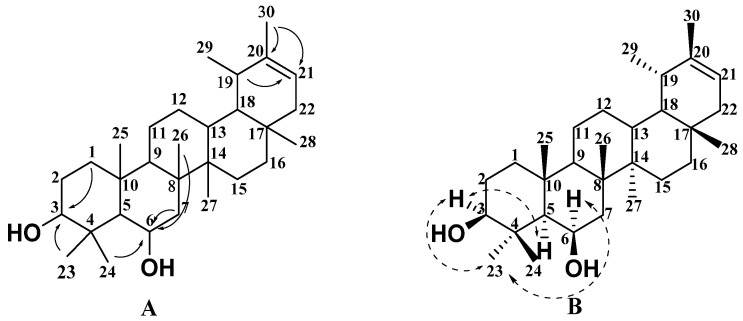
Key HMBC (plain arrow; (**A**)) and NOESY (dash arrow; (**B**)) correlations of compound **8**.

**Figure 5 molecules-27-02161-f005:**
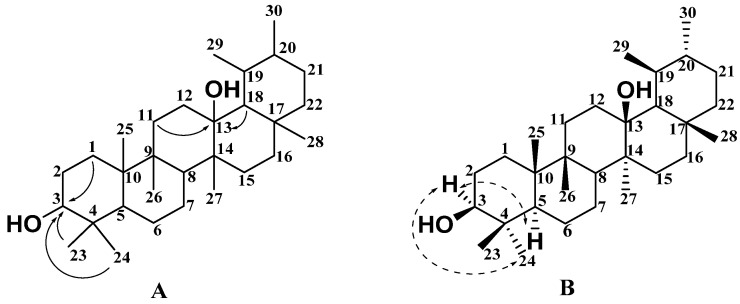
Key HMBC (plain arrow; (**A**)) and NOESY (dash arrow; (**B**)) correlations of compound **9**.

**Figure 6 molecules-27-02161-f006:**
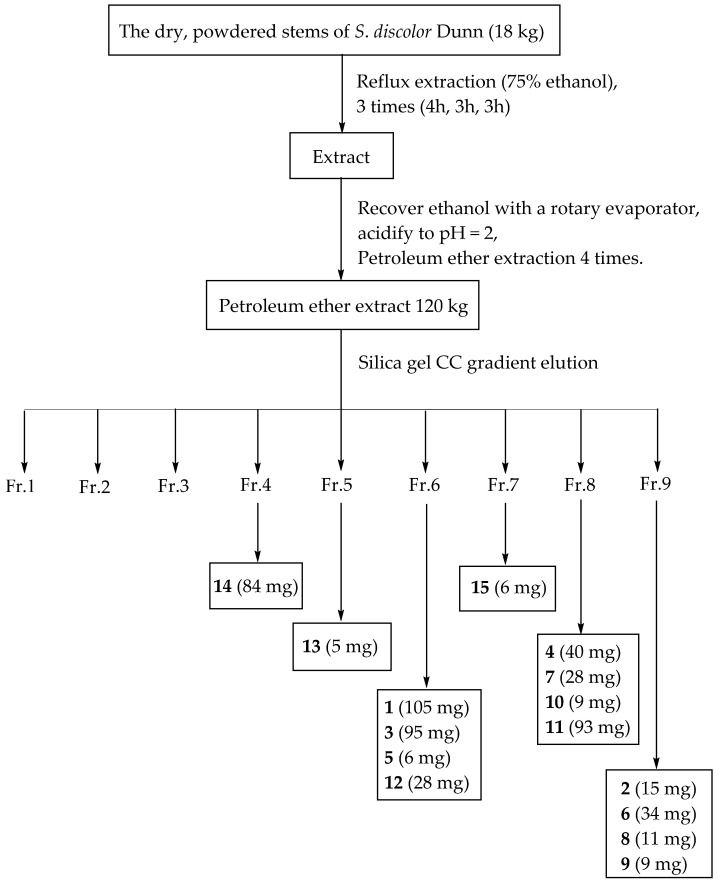
Separation process of chemical constituents from *S. discolor* Dunn.

**Table 1 molecules-27-02161-t001:** ^1^H NMR (600 MHz) data for **1**, **4**, **7**, **8**, and **9** (*δ* in ppm and *J* in Hz).

Position	1 ^a^	4 ^a^	7 ^b^	8 ^a^	9 ^b^
1			3.80 (dd, 11.1, 4.2)	1.85 (m) 1.63 (t, 1.7)	1.22 (s) 1.42 (s)
2	3.23 (d, 10.9) 3.30 (d, 10.9)	3.57 (d, 10.9) 3.23 (d, 10.9)	2.34 (q, 12.0) 2.42 (dt, 12.6)	2.04 (m) 1.74 (m)	1.01 (s) 1.75 (d, 4.6)
3	3.86 (dd, 12.0, 4.6)	3.87 (dd, 12.0, 4.6)	3.63 (dd, 12.0, 4.4)	3.48 (dd, 12.4, 4.3)	3.49 (dd, 10.6, 5.6)
5	1.59 (d, 2.8)	2.0 (d, 4.3)	1.80 (d, 3.2)	0.59 (dd, 11.8, 2.2)	0.84 (m)
6	1.51 (d, 4.0) 1.59 (d, 2.9)	2.25 (d, 5.6) 2.0 (d, 4.3)	1.76 (d, 3.0) 1.68 (d, 1.8)	3.26 (dd, 12.4, 4.3)	1.60 (dt, 10.4, 2.6) 1.19 (s)
7	1.35 (d, 3.2) 1.53 (d, 4.2)	1.35 (d, 3.2) 1.53 (d, 4.2)	1.76 (d, 3.0) 1.68 (d, 1.8)	1.52 (d, 3.2) 1.57 (s)	1.30 (d, 3.8) 1.42 (s)
8					1.23 (s)
9	2.30 (dd, 11.3, 5.6)	2.25 (dd, 11.3, 5.6)	2.42 (dd, 12.6, 4.5)	1.52 (d, 3.2)	
11	2.40 (dd, 12.0, 4.8) 2.25 (dd, 11.3, 5.6)	2.40 (dd, 12.0, 4.8) 2.25 (dd, 11.3, 5.6)	1.68 (t, 1.8) 1.80 (d, 3.2)	1.43 (d, 2.9) 2.35 (m)	
12	5.20 (dd, 4.5, 2.9)	5.22 (dd, 4.4, 2.7)	1.80 (d, 3.2) 1.75 (d, 3.0)	1.34 (s) 1.43 (d, 2.9)	1.71 (m) 1.90 (m)
13			1.62 (s)	1.57 (s)	
15	0.98 (s) 1.70 (d, 4.6)	1.01 (s) 1.32 (d, 4.6)	1.68 (t, 1.8) 1.28 (s)	1.34 (s) 1.63 (d, 1.7)	1.23 (s) 1.90 (m)
16	1.19 (s) 1.90 (d, 4.5)	1.21 (s) 2.0 (d, 4.5)	1.68 (t, 1.8) 1.28 (s)	1.26 (s)	1.81 (t, 3.3) 2.15 (d, 3.8)
18	1.97 (dd, 13.5, 4.2)	2.10 (dd, 13.6, 4.3)	1.28 (s)	1.06 (s)	1.33 (s)
19	1.14 (s) 1.74 (s)	0.91 (s) 1.32 (s)	1.96 (s)	1.65 (s)	1.41(s)
20					2.44 (td, 7.3, 3.9)
21	1.17 (d, 2.2) 1.31 (s)	1.17 (d, 2.2) 1.02 (s)	5.33 (dd, 6.7, 2.0)	5.30 (s)	1.79 (d, 3.2) 2.11 (m)
22	1.37 (d, 3.7) 1.53 (d, 4.2)	1.35 (d, 3.7) 1.51 (d, 4.3)	1.62 (s) 1.28 (s)	1.57 (s) 1.74 (m)	1.40 (s) 1.19 (s)
23	*β* 1.10 (s)	*β* 1.08 (s)	*β* 1.24 (s)	*α* 0.93 (s)	*β* 1.27 (s)
24	*α* 1.03 (s)	*α* 1.04 (s)	*α* 1.09 (s)	*β* 0.96 (s)	*α* 0.90 (s)
25	*β* 1.32 (s)	*β* 1.32 (s)	*β* 1.26 (s)	*β* 0.96 (s)	*β* 1.08 (s)
26	*β* 1.00 (s)	*β* 1.02 (s)	*β* 1.17 (s)	*β* 1.06 (s)	*β* 1.33 (s)
27	*α* 1.21 (s)	*α* 1.21 (s)	*α* 1.01 (s)	*α* 0.76 (s)	*α* 1.42 (s)
28	3.30 (d, 3.2) 3.21 (d, 10.9)	3.57 (d, 3.2) 3.23 (d, 10.9)	*β* 0.80 (s)	*β* 0.74 (s)	*β* 1.01 (s)
29	*α* 0.90 (s)	*α* 0.91 (s)	*α* 1.65 (s)	*α* 0.99 (d, 6.4)	*β* 1.03 (d, 2.8)
30	*β* 0.88 (s)	*β* 0.89 (s)	*β* 0.99 (s)	*β* 1.57 (s)	*α* 1.03 (d, 2.8)

^a^ Data measured in CDCl_3_. ^b^ Data measured in C_5_D_5_N.

**Table 2 molecules-27-02161-t002:** ^13^C NMR (150 MHz) data for **1, 4**, **7**, **8**, and **9** (*δ* in ppm).

Position	1 ^a^	4 ^a^	7 ^b^	8 ^a^	9 ^b^
1	214.6	212.4	79.7	38.1	38.5
2	42.8	44.1	39.5	27.1	26.7
3	79.3	78.6	75.5	79.3	77.9
4	38.0	39.3	39.7	38.8	39.2
5	51.3	54	52.1	53.0	55.5
6	18.4	17.8	18.5	75.0	18.6
7	32.3	32.5	34.5	17.9	40.5
8	41.9	42	42.5	41.6	47.9
9	38.9	39.1	53.8	51.4	42.2
10	51.9	52.3	44.1	43.4	37.0
11	25.3	25.3	24.8	24.4	21.3
12	122.9	123	28.4	34.0	38.9
13	143.1	143.1	39.3	36.3	73.8
14	39.9	39.7	42.7	42.4	38.9
15	25.4	25.5	27.5	27.7	28.1
16	21.9	22.0	37.0	36.7	38.3
17	36.9	37	34.8	34.3	35.5
18	42.5	42.5	48.9	48.6	49.6
19	46.1	46.1	36.5	38.3	43.0
20	30.9	30.9	140.0	139.9	41.3
21	34.1	34.1	119.3	118.8	28.6
22	31.0	31.0	42.0	42.2	34.5
23	22.3	16	28.7	12.0	28.4
24	27	28.5	16.2	27.8	16.2
25	15	15	13.2	14.6	16.2
26	17.5	17.5	16.8	16.3	21.6
27	25.8	25.7	14.9	15.0	17.8
28	69.7	69.9	18.0	17.7	18.4
29	33.2	33.2	22.6	22.4	15.9
30	23.5	23.5	21.8	21.6	14.7

^a^ Data measured in CDCl_3_. ^b^ Data measured in C_5_D_5_N.

**Table 3 molecules-27-02161-t003:** *α*-glucosidase inhibitory activity of compounds **1, 3, 7, 8, 9, 13** and **15**. (*n* = 3) ^a^.

Compound	IC_50_ (μM)	Compound	IC_50_ (μM)
**1**	0.27 ± 0.0499	**9**	0.23 ± 0.0307
**3**	0.11 ± 0.0222	**13**	0.26 ± 0.0383
**7**	0.56 ± 0.0331	**15**	0.09 ± 0.0045
**8**	0.23 ± 0.0135	Acarbose ^b^	0.35 ± 0.0006

^a^ Data of inactive compounds are not listed. ^b^ Positive control.

## Data Availability

The data presented in this study are available in the Appendix A.

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
