# Peer review of "Pentacyclic Triterpenoids from Sabia discolor Dunn and Their α-Glycosidase Inhibitory Activities"

_molecules, 2022, doi:10.3390/molecules27072161_

Round 1

Reviewer 1 Report

The manuscript of Ma et al describes the isolation of pentacyclic triterpenoids from Sabia discolor Dunn stems, and the structure identification of 15 compounds with full spectra analysis and structure elucidation of only the four new compounds, 1,7,8, and 9. The author identified the biological activates of the isolated compounds by investigating their α-glycosidase inhibition activities. This study is well designed and I enjoyed reading the structure elucidation part in the results. Although the authors did well in the results part, other sections of the manuscript need more attention especially the material and methods part. Here are some notes:   

Introduction

Line 33:  “Chemotherapy” is very confusing here, replace with another description

Lines 34-36: What is the source (reference) of this information? Synthetic drugs do not, commonly suffer from low bioavailability, and the immediate release means rapid onset time which is normally an advantage. These “disadvantages” are not actual and do not go with the argument for the presence of natural products remedies. This part needs adjustment and rephrasing

Line 41: “α-Glucosidase inhibitors are a new class” actually they are not new anymore, rephrase

Lines 51-53: In this part of the introduction, the results should not be revealed, this part represents the aim of the study. These lines need adjustment

Results

Line 58: What are the parameters of the HPLC process? The column, the flow rate, the detection type…, specify in the material and methods relevant section.

Figure 2: I recommend adding at least the NOESY correlation of compound 4 together with the already present compound 1 so that it will be more obvious how did you conclude the stereo-configuration of the 3 OH of compound 1.

In table 1, if compounds 1 and 4 are isomers and only the 3 OH is the difference, why does compound 4 show fewer 1HNMR signals than compound 1? Furthermore, why does proton 3 in compound 4 give a multiple?

The supplementary figures were not referred to in the text in most cases; please refer to them where relevant, for example, you always refer to the DEPT analysis, however, it is only found in the supplementary

Section 2,2: why did the authors choose to investigate the α-glycosidase inhibition activities of these compounds? Although there is no structural similarity between them and the positive control?

Discussion

All the discussion sections need more attention and need parts to be added. For example, the discussion of the plant’s medicinal activities has no references. Are there any compounds that were isolated before from this plant? What is the interpretation of the authors for compound 7 to have more α-glycosidase inhibition activity than acarbose? And if there are any similar cases in the literature to enrich the discussion.

Materials and methods

Line 193: Section 3.1. General experimental procedures: This section does not describe experimental procedures, these are just the materials and some of the experimental conditions used. Change the section heading to fit.

Section 3.3 Lines 218-249: all the following needs to be inserted in the manuscript if relevant:

You used 18 kg of the dried stems and reflux them with 75% ethanol for 10 hours as a total. First, 18 kg is a massive weight for extraction, how did you reflux such a weight? And I think 10 hours are very less, this amount of the plant needs more time even for reflux

Why did you use reflux? And you did not use normal maceration with the alcohol especially since you really do not know if there are thermolabile compounds in the plant or not

Why stem only, why not use leaves or the whole herb?

What is the temperature of the reflux and was it under normal pressure?

The ethanol was recovered, how? Was it under reduced pressure?

What is the weight of the total extract you got from the total extraction process?

What is the meaning of (pH = 2) in line 221?

In line 222: it is not clear how you did the solvent gradient? Was it over a silica gel column? HPLC? And if HPLC, what are the other parameters of the process i.e. the flow rate, the column type, ….

The rest part of the section is well described and I appreciate the amount of effort exerted here to obtain all these compounds, however, I recommend making a hierarchy figure to illustrate these fractions and sub-fractions and from which fraction did the compound emerged. It will be easier for the reader to follow.

Section 3.4: What is the positive control you used? And what is the meaning of background group?   

Author Response

Response to reviewer 1

According to reviewer 1’s comments, our manuscript has been revised. The modification traces were marked with red front in our revised manuscript, and response to reviewers is as the following:

For the “Reviewer #1”

Introduction

  1. Reviewer: Line 33: “Chemotherapy” is very confusing here, replace with another description.

Response: According to reviewer’s suggestion, “Chemotherapy” in Line 33 has been replaced with “Long-term medication” in our revised manuscript.

  1. Reviewer: Lines 34-36: What is the source (reference) of this information? Synthetic drugs do not, commonly suffer from low bioavailability, and the immediate release means rapid onset time which is normally an advantage. These “disadvantages” are not actual and do not go with the argument for the presence of natural products remedies. This part needs adjustment and rephrasing.

Response: According to reviewer’s suggestion, we have adjusted and revised the corresponding description lines 34-36 of our manuscript.

  1. Reviewer: Line 41: “α-Glucosidase inhibitors are a new class” actually they are not new anymore, rephrase.

Response: According to reviewer’s suggestion, “α-Glucosidase inhibitors are a new class” in Line 41 has been revised as “α-Glucosidase inhibitors are an important class”.

  1. Reviewer: Lines 51-53: In this part of the introduction, the results should not be revealed, this part represents the aim of the study. These lines need adjustment.

Response: According to reviewer’s suggestion, we have revised the corresponding description which represented the aim of the study in the part of introduction.

  1. Reviewer: Line 58: What are the parameters of the HPLC process? The column, the flow rate, the detection type…, specify in the material and methods relevant section.

Response: According to reviewer’s suggestion, we have added parameters of the HPLC process as shown in the material and methods relevant section.

  1. Reviewer: Figure 2: I recommend adding at least the NOESY correlation of compound 4 together with the already present compound 1 so that it will be more obvious how did you conclude the stereo-configuration of the 3 OH of compound 1.

Response: According to reviewer’s suggestion, we have added the NOESY profile of compound 4 and the NOESY correlation of compound 4 together with the already present compound 1 in figure 2.

  1. Reviewer: In table 1, if compounds 1 and 4 are isomers and only the 3 OH is the difference, why does compound 4 show fewer 1HNMR signals than compound 1? Furthermore, why does proton 3 in compound 4 give a multiple?

Response: According to reviewer’s suggestion, we have corrected and added proton signals at compound 4 in table 1. Thank you very much for your suggestion.

  1. Reviewer: The supplementary figures were not referred to in the text in most cases; please refer to them where relevant, for example, you always refer to the DEPT analysis, however, it is only found in the supplementary.

Response: According to reviewer’s suggestion, we have added supplementary figures in relevant parts of our revised manuscript.

  1. Reviewer: Section 2,2: why did the authors choose to investigate the α-glycosidase inhibition activities of these compounds? Although there is no structural similarity between them and the positive control?

Response: First of all, we found that pentacyclic triterpenoids have a certain effect on T2DM through literature review [ Biomedecine & pharmacotherapie, 2017, 94:726-741.]. α-glucosidase inhibitors are an important class of drugs that can be used for the treatment of T2DM and widely exist in the community fruits, leaves, seeds and other tissues and organs of plants. Until now, no positive control substance similar to pentacyclic triterpene structure was found, and acarbose has good efficacy, which was recognized as a positive control for the detection of α -glycosidase inhibition rate in current scientific research.

  1. Reviewer: All the discussion sections need more attention and need parts to be added. For example, the discussion of the plant’s medicinal activities has no references. Are there any compounds that were isolated before from this plant? What is the interpretation of the authors for compound 7 to have more α-glycosidase inhibition activity than acarbose? And if there are any similar cases in the literature to enrich the discussion.

Response: Before our research, we have conducted systematic Literature research, and it was found that there were triterpenes, alkaloids, chain aliphatic hydrocarbons and other compounds in other plants of the genus Sabia, and there were no reports on the chemical composition of Sabia discolor Dunn. According to reviewer’s suggestion, we have added relevant content and references in the discussion section. We are very sorry that the description of the active part of compound 7 was incorrect due to our carelessness, and we have corrected the corresponding section of the manuscript.

  1. Reviewer: Line 193: Section 3.1. General experimental procedures: This section does not describe experimental procedures, these are just the materials and some of the experimental conditions used. Change the section heading to fit.

Response: According to reviewer’s suggestion, “General experimental procedures” in Section 3.1 has been revised as “General experimental materials”.

  1. Reviewer: Section 3.3 Lines 218-249: all the following needs to be inserted in the manuscript if relevant:

(1). “You used 18 kg of the dried stems and reflux them with 75% ethanol for 10 hours as a total. First, 18 kg is a massive weight for extraction, how did you reflux such a weight? And I think 10 hours are very less, this amount of the plant needs more time even for reflux.”

Response: We possessed a 300L extraction tank (the model is presented in section 3.1), which can realize the one-time extraction of material weighing up to 50 kg, so the extraction task of 18 kg sample in this project can be realized. According to literature research, it was found that the reflux time of about 10 hours is a common extraction process, so we also adopted this method.

(2). Why did you use reflux? And you did not use normal maceration with the alcohol especially since you really do not know if there are thermolabile compounds in the plant or not.”

Response: Preliminary research was found that reflux extraction was the most common method of intractable extraction in phytochemistry, so we chose the commonly used reflux extraction method, which was impregnated by ethanol before extraction. In addition, the temperature in this extraction method does not affect the pentacyclic triterpenoids we got.

(3). “Why stem only, why not use leaves or the whole herb?”

Response: As we all know, choosing the whole plant as research object was destructive. According to pre-experiments, it was found that secondary metabolites in the stems were more diverse than those of the leaves by thin layer chromatography, so we used the stems of Sabia Discolor Dunn as the research object.

(4). “What is the temperature of the reflux and was it under normal pressure?”

Response: Reflux occurs under normal pressure at 79.7 °C.

(5). “The ethanol was recovered, how? Was it under reduced pressure?”

Response: We The ethanol was recovered under reduced pressure by rotating the evaporator.

(6). “What is the weight of the total extract you got from the total extraction process?”

Response: In our manuscript, we used 75% ethanol to extract this plant, the extract was treated by recovering ethanol to obtain an aqueous solution. In the whole process, we did not get the total amount of extraction, but to get the weight of the individual extracts as quickly as possible. We only studied the extraction site of petroleum ether, and obtained the petroleum ether extract of 120 g after concentration.

(7). “What is the meaning of (pH = 2) in line 221?”

Response: “pH = 2” means to acidify the extract to the value of pH=2. In the course of pre-experiment, it was found that Sabia discolor Dunn might contain some alkaloids by the detecting of alkaloid developer, so we used the acid to deal with the ethanol extract to achieve non-alkaloids and in order to be more clearly described, we have added the separation process and Figure 6 in our revised manuscript.

(8). “In line 222: it is not clear how you did the solvent gradient? Was it over a silica gel column? HPLC? And if HPLC, what are the other parameters of the process i.e. the flow rate, the column type, ….”

Response: We used silica gel column chromatography for gradient elution in the manuscript.

(9). “The rest part of the section is well described and I appreciate the amount of effort exerted here to obtain all these compounds, however, I recommend making a hierarchy figure to illustrate these fractions and sub-fractions and from which fraction did the compound emerged. It will be easier for the reader to follow.”

Response: According to reviewer’s suggestion, we have added a hierarchy figure as shown in Figure 6 to illustrate these fractions and sub-fractions and from which fraction did the compound emerged.

  1. Reviewer: Section 3.4: What is the positive control you used? And what is the meaning of background group?

Response: The positive control we used is Acarbose. The background group was a control group that did not add positive drugs and α-glycosidase inhibitors. And we have added relevant content in section 3.4.

Your kindly consideration and your advice will be highly appreciated. If I can be of any assistance regarding the process of our manuscript, please feel free to contact me. I am looking forward to hearing from you soon.

Thank you very much for your suggestion and help.

Best regards,

Shu-Zhen Mu

State Key Laboratory of Functions and Applications of Medicinal Plants, Guizhou Medical University

No. 3491, Baijin Road, Guiyang 550014, Guizhou, P. R. China

Tel: +86-851-83802214; Fax: +86-851-83805081

E-mail: muzi0558@126.com

Reviewer 2 Report

This manuscript describes the isolation and structural determination of pentacyclic triterpenoids from ethnic medicinal plant Sabia discolor Dunn. The authors found four new triterpenoid compounds. The authors also evaluated the inhibitory activities of isolated compounds against α-glycosidase. The obtained information might be useful in the field. There are minor suggestions.

  1. section 2.3: The authors could discuss about the structure-activity relationship in more detail. For example, it could be mentioned which functional groups are important for the inhibition activity.
  2. line 184-185 and 271-272: It is described that the new compounds 7 has more significant activity than the positive control (Acarbose). However, IC50 value of acarbose is lower than that of compound 7 in Table 3, suggesting that acarbose has stronger inhibition activity than the compound 7.
  3. Table 1: Some of the proton signals are not assigned in Table 1 (e.g. positions 6, 7, 9 for compound 7). All signals should be assigned.
  4. It should be mentioned which α-glucosidase was used in this study. The only described thing is that α-glucosidase was obtained from Sigma Company (USA) (line 209-210). Is it α-glucosidase from Saccharomyces cerevisiae or from Bacillus stearothermophilus?
  5. line 220-221 "The above extract (pH 2)": Was the above extract acidified?

Author Response

Response to reviewer 2

According to reviewer 2’s comments, our manuscript has been revised. The modification traces were marked with red front in our revised manuscript, and response to reviewers is as the following:

For the “Reviewer #2”

  1. Reviewer: section 2.3: The authors could discuss about the structure-activity relationship in more detail. For example, it could be mentioned which functional groups are important for the inhibition activity.

Response: According to reviewer's suggestion, we have added a brief description of the relevant structure-activity relationship in section 3.2.

  1. Reviewer: line 184-185 and 271-272: It is described that the new compounds 7 has more significant activity than the positive control (Acarbose). However, IC50 value of acarbose is lower than that of compound 7 in Table 3, suggesting that acarbose has stronger inhibition activity than the compound 7.

Response: Thanks for reviewer’s reminder, we have corrected the description of the activity of compound 7 in our revised manuscript.

  1. Reviewer: Table 1: Some of the proton signals are not assigned in Table 1 (e.g. positions 6, 7, 9 for compound 7). All signals should be assigned.

Response: According to reviewer’s suggestion, we have assigned signals for all protons in Table 1 of our revised manuscript.

  1. Reviewer: It should be mentioned which α-glucosidase was used in this study. The only described thing is that α-glucosidase was obtained from Sigma Company (USA) (line 209-210). Is it α-glucosidase from Saccharomyces cerevisiae or from Bacillus stearothermophilus?

Response: According to reviewer’s suggestion, we have added the source of α-glycosidase in section 3.1.

  1. Reviewer: line 220-221 "The above extract (pH 2)": Was the above extract acidified?

Response: Yes, we have acidified the extract for the initial separation of alkaloids and other components. We are very sorry that our description is not clear, which has affected the reviewer's understanding, and now we have revised this part in section 3.3 in our revised manuscript.

Your kindly consideration and your advice will be highly appreciated. If I can be of any assistance regarding the process of our manuscript, please feel free to contact me. I am looking forward to hearing from you soon.

Thank you very much for your suggestion and help.

Best regards,

Shu-Zhen Mu

State Key Laboratory of Functions and Applications of Medicinal Plants, Guizhou Medical University

No. 3491, Baijin Road, Guiyang 550014, Guizhou, P. R. China

Tel: +86-851-83802214; Fax: +86-851-83805081

E-mail: muzi0558@126.com

Reviewer 3 Report

Please express the IC50 in μM. 

Please provide in the manuscript complete NMR data for all new compounds. Without these data, the results sound equivocal.  

Author Response

Response to reviewer 3

According to reviewer 3’s comments, our manuscript has been revised. The modification traces were marked with red front in our revised manuscript, and response to reviewers is as the following:

For the “Reviewer #3”

  1. Reviewer: Please express the IC50 in μM.

Response: Thanks for reviewer’s reminder, and we are sorry for using the wrong unit because of our carelessness, we have modified the IC50 value units as shown in Table 3.

  1. Reviewer: Please provide in the manuscript complete NMR data for all new compounds. Without these data, the results sound equivocal.

Response: According to reviewer’s suggestion, we have added NMR data for all new compounds in our revised manuscript.

Your kindly consideration and your advice will be highly appreciated. If I can be of any assistance regarding the process of our manuscript, please feel free to contact me. I am looking forward to hearing from you soon.

Thank you very much for your suggestion and help.

Best regards,

Shu-Zhen Mu

State Key Laboratory of Functions and Applications of Medicinal Plants, Guizhou Medical University

No. 3491, Baijin Road, Guiyang 550014, Guizhou, P. R. China

Tel: +86-851-83802214; Fax: +86-851-83805081

E-mail: muzi0558@126.com

Round 2

Reviewer 1 Report

The manuscript looks better now, however still some points not answered:

Figure 2: Specify what are A, B, and C in the figure legend, especially B and C, which is for compound 1 and compound 4 

the authors did not answer the question: "Section 2,2: why did the authors choose to investigate the α-glycosidase inhibition activities of these compounds? Although there is no structural similarity between them and the positive control?" and the reference chosen in the response is not appropriate. what I mean here is why "α-glycosidase inhibition" to treat diabetes, and why not other mechanisms such as insulin secretagogues, or increase insulin sensitivity or any other mechanisms 

I think the discussion part still needs more attention as I indicated in my last comment

Author Response

According to reviewer 1’s comments, our manuscript has been revised again. The modification traces were marked with color front in our revised manuscript, and response to reviewers is as the following:

  1. Reviewer: Figure 2: Specify what are A, B, and C in the figure legend, especially B and C, which is for compound 1 and compound 4.

Response: Thanks for reviewer’s reminder, and we are sorry for unclear expression because of our carelessness, we have corrected the related description and specified A, B and C in figure 2, and also specified A and B in figures 3 ‒ 5.

  1. Reviewer: the authors did not answer the question: "Section 2,2: why did the authors choose to investigate the α-glycosidase inhibition activities of these compounds? Although there is no structural similarity between them and the positive control?" and the reference chosen in the response is not appropriate. what I mean here is why "α-glycosidase inhibition" to treat diabetes, and why not other mechanisms such as insulin secretagogues, or increase insulin sensitivity or any other mechanisms.

Response: Thanks for reviewer’s advice, and we are sorry that we didn't fully understand your question before. According to literatures ([1]‒[3]) as shown in the following, we found that the inhibition rate of pentacyclic triterpenoids on α-glycosidase have been reported. So we chose to investigate the α-glycosidase inhibition activities of our compounds. And the question about “why is α-glycosidase inhibition to treat diabetes” is not our research focus in our manuscript. In our manuscript, some pentacyclic triterpenoids were obtained and their α-glycosidase inhibition activities were evaluated, and their activities are closely related to diabetes, which provided a material basis for the research and development of diabetes drugs. And we think, “further mechanisms such as insulin secretagogues, or increase insulin sensitivity or any other mechanisms” are not the focus of our manuscript.

[1] Chen L, Yang X S, Yang J, et al. Synthesis and inhibitory activity against α-glycosidase of pentacyclic triterpenes[J]. Journal of China Pharmaceutical University, 2010, 41(3):222-225.

[2] Bo-wei Zhang, Yan Xing, Chen Wen, et al. Pentacyclic triterpenes as α-glucosidase and α-amylase inhibitors: Structure-activity relationships and the synergism with acarbose.[J]. Bioorganic & Medicinal Chemistry Letters, 2017, 27(22): 5065-5070.

[3] Q Yu, J Qi, L Wang, et al. Pentacyclic Triterpenoids from Spikes ofPrunella vulgarisL. Inhibit Glycogen Phosphorylase and Improve Insulin Sensitivity in 3T3-L1 Adipocytes[J]. Phytotherapy Research, 2014.

  1. Reviewer: I think the discussion part still needs more attention as I indicated in my last comment.

Response: Thanks for reviewer’s advice. According last comment, we have added some description in the discussion part as shown in last revised manuscript. But by searching Scifinder database, no relevant literature is directly reported the plant (“Sabia discolor Dunn”, and the search results are shown in the following. In other words, we are the first to study this plant systematically. So we have added the discussion related to our findings.

Thank you very much for your suggestion and help.

Best regards,

Shu-Zhen Mu

Reviewer 2 Report

The authors have adequately revised the manuscript according to my comments.

Author Response

Thanks for your advices.